# Artificial Intelligence-Enhanced UUV Actuator Control

**Zhiyu Wang [1] and Timothy Sands [2,*]**

[1] Sibley School of Mechanical and Aerospace Engineering, Cornell University, Ithaca, NY 14853, USA
[2] Department of Mechanical Engineering (CVN), Columbia University, New York, NY 10027, USA
* Correspondence: dr.timsands@caa.columbia.edu

**Abstract:** This manuscript compares deterministic artificial intelligence to a model-following control applied to DC motor control, including an evaluation of the threshold computation rate to let unmanned underwater vehicles correctly follow the challenging discontinuous square wave command signal. The approaches presented in the main text are validated by simulations in MATLAB®, where the motor process is discretized at multiple step sizes, which is inversely proportional to the computation rate. Performance is compared to canonical benchmarks that are evaluated by the error mean and standard deviation. With a large step size, discrete deterministic artificial intelligence shows a larger error mean than the model-following self-turning regulator approach (the selected benchmark). However, the performance improves with a decreasing step size. The error mean is close to the continuous deterministic artificial intelligence when the step size is reduced to 0.2 s, which means that the computation rate and the sampling period restrict discrete deterministic artificial intelligence. In that case, continuous deterministic artificial intelligence is the most feasible and reliable selection for future applications on unmanned underwater vehicles, since it is superior to all the approaches investigated at multiple computation rates.

**Keywords:** unmanned underwater vehicle (UUV); deterministic artificial intelligence; model-following; recursive least squares; marine actuators; self-tuning regulators



## 1. Introduction

### 1.1. Motivation

Over the past decades, increasing threats and emerging situations have posed a significant challenge to the coast guard worldwide. Navies are moving to innovate and adapt new technology to build a more lethal and distributed naval force for the future [1]. The United States seems to tackle the problem using unmanned technologies and applications. Unmanned underwater vehicles seem to be one of the reliable solutions for successful maritime interdiction missions [2], emphasizing the automation of DC motor control.

The vehicle is controlled by the fin-shape actuator shown in Figure 1a,b. The unmanned underwater vehicle can sail along the disparate trajectory by sending a command signal to the motors connected to the actuator (e.g., Figure 1). In this manuscript, the efficacy of motor control techniques is evaluated with respect to system discretization.

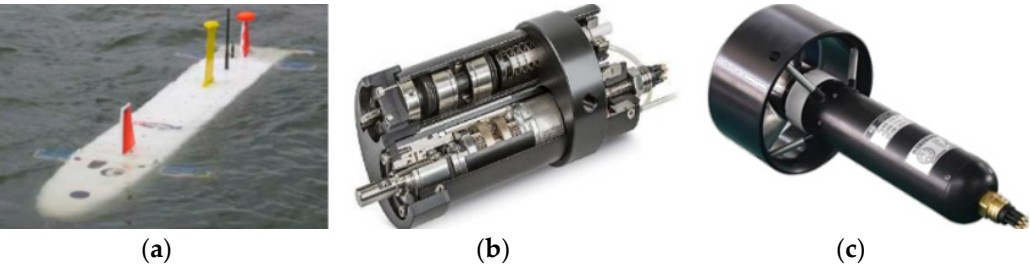

**Figure 1.** (**a**) Aries unmanned underwater vehicle [3]; (**b**) ECI-40 Maxon underwater drive motor and gear head [4]; and (**c**) underwater thruster propeller motor [5].

## 1.2. Literature Review

Sarton's formulation of autopilot techniques used disparate thrust for an Aries autonomous underwater vehicle in 2003 [6], while Tan focused on application of horizontal steering control during docking [7]. Sarton sought advantages in differential thrusts to counter wind and wave disturbances while surfaced, by creating different motor voltages supplied to each propeller motor. Seeking autonomous recovery or docking operations, Tan focused on the capability of the vehicle to track and steer itself accurately despite constant perturbation by wave motion effects necessitating an accurate acoustic homing during the final stages of the docking with high update rates of the acoustic systems. The Aries vehicle was experimentally tested to have an update rate of only about 0.3 Hz. These delayed data can potentially cause a false commanded reference input to the tracking system in between the updates and can cause Aries to miss the moving cage's entrance motivating high-interest investigating systems that prove efficacy at such low data rates. More recently, Wang et al. [8] proposed a fault-tolerant control based on nonlinear programming, motivating the investigation of efficacy at low data rates since such methods ubiquitously require increased computational times.

Dinç proposed a least squares parameter identification to estimate the hydrodynamic coefficients for autonomous underwater vehicles in the presence of measurement biases [9], while this present manuscript illustrates the application of such techniques to the estimation of actuator motor parameters. Gutnik et al. demonstrated how the efficacy of such approaches manifests in operational capabilities for autonomous, near-seabed visual imaging missions [10], particularly regarding thrust allocation for path following.

The development of adaptive and learning systems has a long history that is presently culminating in a very recent and distinguished lineage in the literature [4,11–14] whose results are presented in Figure 2, already bestowing three recent major publication awards [15,16], validating the contemporary interest in continued developments. Many techniques are available from this long lineage as alternative benchmarks for comparing newly proposed methods. Rathmore focused on artificial intelligence, applying a robust steering control based on tuning of the PID controller using the so-called genetic algorithm and the harmonic search algorithm [11]. In 2020, deterministic artificial intelligence was proposed to control autonomous unmanned underwater vehicles [12].

*The proposed method is based on feedforward self-awareness using process dynamics.*

Koo formatted the feedback signal as an adaptive and learning system (proportional derivative feedback and two-norm optimal least squares) [13]. As a duplication and continuation of Koo's work in [13], this manuscript provides an in-depth comparison between the performance of the deterministic artificial intelligence and declared benchmark approaches with iterated step sizes. The scope of this sequel is exactly that proposed as future research by Koo in 2022. The performance evaluation mainly focuses on the ability to track the command signals represented by challenging square wave trajectories.

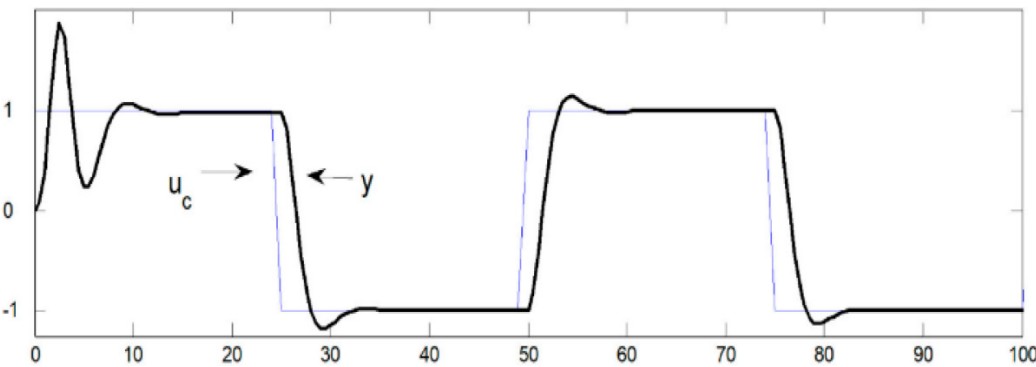

**Figure 2.** Comparative benchmark [13,14] used in the lineage of this research's existing related work.

Bernat introduced a speed control approach [17] based on a model-reference adaptive control algorithm for torque load and ripple compensation. Gowri described direct torque control as one approach focused on discontinuities in rapid modulating commands [18]. Moreover, Rathaiah [19] and Haghi [20] both proposed extremum-seeking adaptive control of first-order systems, which is similar to the approach applied to the vehicle automation control described in [21]. The methods are similarly tested and compared with discontinuous step and square wave inputs. For DC motors, tracking the discontinuous step functions and square wave functions is challenging since overshoot occurs at the square wave's discontinuities, significantly influencing the tracking performance of nonlinear adaptive control approaches. The difficulty is validated by Vidlak's very contemporary example in Figures 24 and 26 in [22]. This facet is also elaborated by Vidlak's work on tracking the performance of self-turning regulators [22] and model-reference adaptive control [23].

A similar phenomenon is revealed in Section 3 of this manuscript for the error distribution comparison of different algorithms. This manuscript describes a duplication and continuation of the work presented in [13], which is also work based on its prequel research by Shah [14], where model-following self-turning regulators [24] are chosen as the comparative benchmark approach.

The learning approach evaluated in this manuscript (displayed in Figure 3) stems from Slotine [25] and the nonlinear adaptive method for robotics [26]. The technique was quickly transformed to an expression in the coordinates of the body reference frame [27]. In [28], the tunability of the feedback and feedforward elements are demonstrated, substantiating the self-awareness statements of deterministic artificial intelligence [29]. As described by Fossen in [30], alternative trajectory tracking control mechanisms based on classical proportional, integral, and derivative control; a linear-quadratic regulator; feedback linearization; nonlinear backstepping; and sliding mode control are presented. Some approaches are applied to the ocean vehicle mentioned in [31]. The efficacy of such systems is also simulated in [32]. Lorenz and his students also used and evaluated the feedforward element in [33–40].

*Feedforward self-awareness using process dynamics is foundational for the proposed method.*

The fault-tolerance [35], loss reduction [36], loss minimization [37], dead-beat control [38], and self-sensing [22,39] are evaluated and extended from the vehicle to the actuator control circuit. As described in [21,22], the precursor of using the physics-based dynamics for virtual sensing, which follows the illustration of optimality and self-sensing, is applied explicitly to DC motors.

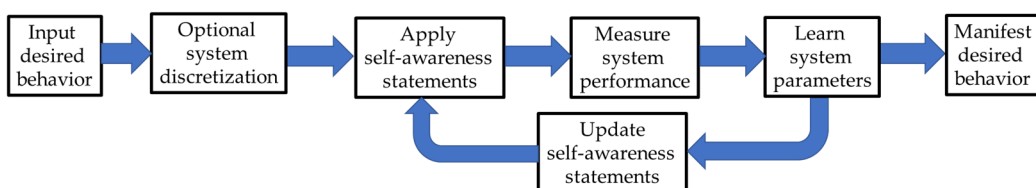

**Figure 3.** Flow chart qualitatively elaborating deterministic artificial intelligence.

### 1.3. Research Gap and Authors Contribution

This manuscript continues the investigation of Koo [13] using a deterministic artificial intelligence learning approach following the recommendation provided by Shah [14]. In Section 3, the comparison analysis of the computational rate is evaluated for the adaptive and learning methods. More step sizes are applied to the compared approaches to assess their performance based on the error mean and error standard deviations.

**Main Conclusion of the study**

As a sequel to Koo [13], multiple propositions are evaluated in Section 2. Validating research is included in Section 3, with direct comparisons between each algorithm based on canonical figures of merit.

1. Recommendation of key threshold discretization and computational speed to duplicate the results of the original prequel [3].
2. Validation of the first sequel's [14] identification of paramountcy of discretization and computational speed.
3. Validation of the second sequel's [13] identification of deterministic artificial intelligence performance and recommended selection of algorithms.

### 1.4. Organization

Section 2 describes the research methods and materials used. The discussion opens with an explanation of the model used as an assumed truth. Next, the comparative benchmark, model-following control is explained and is followed by the proposed deterministic artificial intelligence method. Section three includes the results of a direct performance comparison using simulation experiments investigating the continuous and discrete forms measured by several canonical figures of merit (e.g., means and standard deviations of tracking errors).

## 2. Materials and Methods

This section includes sufficient detail for others to replicate the published result. A truth model for motor dynamics and a model-following self-tuner are described in Sections 2.1 and 2.2, while the newest method (deterministic artificial intelligence) is elaborated in Section 2.3. Deterministic artificial intelligence is a learning method that first asserts self-awareness and then learns in the context of that self-awareness alone.

### 2.1. Truth Model for Motor Dynamics

As described by [33], consider a continuous-time process and precisely normalized model for a DC motor [13] whose transfer function is described in Equation (1).

$$G(s) = \frac{B(s)}{A(s)} = \frac{1}{s(s+1)} \tag{1}$$

Equation (2) shows the z-transform for the discrete-time signal under the frequency domain. Using the MATLAB® function provided in Appendix A to perform the discretization, a time step of 0.5 s was used as the initial continuous-time process. $U(z)$ and $Y(z)$ represent the control signal and output in discrete-time using the z-transform. The final form of the difference equation is written as Equation (3), where proximal variables and nomenclature and defined in periodic tables (e.g., Table 1 in this instance).

$$G(z) = \frac{Y(z)}{U(z)} = \frac{BT}{AR + BS} = \frac{0.0984z + 0.0984}{z^2 - 1.607z + 0.6065} \tag{2}$$

$$0.0984u(t) + 0.0984u(t-1) = y(t+1) - 1.607y(t) + 0.6065y(t-1) \tag{3}$$

**Table 1.** Table of proximal variables and nomenclature [1].

| Variable/Acronym | Definition | Variable/Acronym | Definition |
| --- | --- | --- | --- |
| $G$ | Transfer function | $s$ | Differential variable |
| $Y$ | Output | $z$ | Difference variable |
| $U$ | Input | $t$ | Discrete time variable |

[1] Such tables are offered throughout the manuscript to aid readability.

### 2.2. Model-Following Self Tuner

As shown in Figure 3.3 in reference [24], a dynamic feedforward is typically applied to the input $u_c$, and negative dynamic feedback is applied on output $y$. In that case, the

process input $u$ is produced. The system can be described in Equations (4) and (5), where $U(z)$ and $Y(z)$ represent the control signal and output under the z-transform domain.

$$Y(z) = \frac{B}{A}\left(\frac{T}{R}U(z) - \frac{S}{R}Y(z)\right) \tag{4}$$

$$G(z) = \frac{Y(z)}{U(z)} = \frac{BT}{AR + BS} \tag{5}$$

Due to the designed topology, some cancellation is available in Equation (5), where $B^+$ represents the canceled zeros, $B^-$ represents uncancelled zeros, which represents a scalar multiple to the system, and $A_0$ represents the pole-zero cancellation used to generate the desired system transfer function [14].

$$\frac{Y(z)}{U(z)} = \frac{B^+ B^- A_0 B'_m}{A_0 A_m B^+} = \frac{BT}{AR + BS} = \frac{B_m}{A_m} \tag{6}$$

$B^+$ is set to 1 since no process zero cancellation should be included in the demonstrated control design. As described in [24], $R$, $S$, and $T$ are in the first order. By manipulations of Equation (6), the coefficient of $R$, $S$, and $T$ can be defined as Equations (7)–(9) in terms of estimated and desired process parameters.

$$r_1 = \frac{b_1}{b_0} + \frac{\left(b_1^2 - a_{m_1} b_0 b_1 + a_{m_2} b_0^2\right)(-b_1 + a_0 b_0)}{b_0\left(b_1^2 - a_1 b_0 b_1 + a_2 b_0^2\right)} \tag{7}$$

$$s_0 = \frac{b_1\left(a_0 a_{m_1} - a_2 - a_{m_1} a_1 + a_1^2 + a_{m_2} - a_1 a_0\right)}{b_1^2 - a_1 b_0 b_1 + a_2 b_0^2} + \frac{b_0\left(a_{m_1} a_2 - a_1 a_2 - a_0 a_{m_2} + a_0 a_2\right)}{b_1^2 - a_1 b_0 b_1 + a_2 b_0^2} \tag{8}$$

$$s_1 = \frac{b_1\left(a_1 a_2 - a_{m_1} a_2 + a_0 a_{m_2} - a_0 a_2\right)}{b_1^2 - a_1 b_0 b_1 + a_2 b_0^2} + \frac{b_0\left(a_2 a_{m_2} - a_2^2 - a_0 a_{m_2} a_1 + a_0 a_2 a_{m_1}\right)}{b_1^2 - a_1 b_0 b_1 + a_2 b_0^2} \tag{9}$$

Then, the control signal fed into the process can be described by Equations (10) and (11) after combining feedforward and feedback.

$$RU(z) = TU_c(z) + SY(z) \tag{10}$$

$$u(t) = \beta U_c(t) + \beta a_0 U_c(t-1) + s_0 Y(t) + s_1 Y(t-1) - r_1 U(t-1) \tag{11}$$

As described in reference [24], $\beta$ drives the system to unity gain, which is essential for attaining zero asymptotic tracking error.

### 2.3. Deterministic Artificial Intelligence

Deterministic artificial intelligence requires self-awareness assertion in the feedforward, which can be established by isolating $u(t)$ in the left-hand side of Equation (3), where proximal variables and nomenclature and defined in periodic tables (e.g., Table 2 in this instance). Duplicating Equations (12)–(14) from reference [13] shows the manipulation to express the $u(t)$ to the product of matrix of knowns that are assembled into the matrix $[\phi_d]$, while unknowns are assembled into the vector $\{\hat{\theta}\}$. The unknown vector, $\{\hat{\theta}\}$, is the learned parameters from feedback that can generate the process input. $u^*(t)$ is the regression form of input $u(t)$.

$$u(t) = \frac{1}{0.0984}y(t+1) - \frac{1.607}{0.0984}y(t) + \frac{0.6065}{0.0984}y(t-1) - u(t-1) \tag{12}$$

$$u^*(t) = \hat{a}_1 y_d(t+1) - \hat{a}_2 y_d(t) + \hat{a}_3 y_d(t-1) - \hat{b}_1 u_d(t-1) \tag{13}$$

$$u^*(t) = [\phi_d]\{\widehat{\theta}\} = [y_d(t+1) - y_d(t) + y_d(t-1) - u_d(t-1)]\begin{Bmatrix} \widehat{a}_1 \\ \widehat{a}_2 \\ \widehat{a}_3 \\ \widehat{b}_1 \end{Bmatrix} \quad (14)$$

**Table 2.** Table of proximal variables and nomenclature [1].

| Variable/Acronym | Definition | Variable/Acronym | Definition |
|---|---|---|---|
| $u^*$ | Control input | $\Phi_d$ | Regressor matrix |
| $y_d$ | Desired output | $\hat{\theta}$ | Parameter vector |
| $\hat{a}_1, \hat{a}_2, \hat{a}_3, \hat{b}_1$ | Estimates | $a_1, a_2, a_3, b_1$ | True values |
| $k_p$ | Proportional gain | $k_d$ | Difference gain |
| $A$ | Coefficients of $U(z)$ | $B$ | Coefficients of $Y(z)$ |

[1] Such tables are offered throughout the manuscript to aid readability.

Apply feedforward control and modify the state form $y(t)$ to $y(t+1)$ in Equation (3). Then, the desired trajectory can be computed. To learn the updated feedback parameters $[\hat{\theta}]$, the rough initial estimates of the feedback parameters along with the values of output $y$ and regression $u^*(t)$ are used in recursive least squares [13]. Convert the transfer function in Equation (1) back into the ordinary differential equation reparametrized as in Equation (13). The continuous system can be evaluated by using the deterministic artificial intelligence method. As described in [29], the optimal feedback adjustment can be applied to the system to learn the feedback parameters in the discrete environment, shown in Equation (15).

$$u \equiv \phi_d \left( \phi_d^T \phi_d \right)^{-1} \phi_d^T \delta u \quad (15)$$

The control input $u(t)$ and sinusoidal trajectory (given by Equation (16)) could be obtained by putting the updated and optimal feedback parameters back into Equation (13). The $A_0$ and $A$ in Equation (15) are the original and target states, respectively.

$$z = (A - A_0)[1 + \sin(\omega t + \phi)] \quad (16)$$

## 3. Results

This section shows the comparison between the model-following self-tuner control in Equation (11) and the discrete deterministic artificial intelligence in Equation (15) from Section 3. Compared with the resultant in [13], a larger step size is applied to validate the statement mentioned by Koo: *discrete deterministic artificial intelligence is more susceptible to larger step sizes but more efficacious with a smaller step size* [13]. Moreover, the comparison between continuous and discrete deterministic artificial intelligence is shown in Section 3.2.

### 3.1. Model-Following Self-Tuner Control vs. Deterministic Artificial Intelligence

Comparing the error mean and standard deviation under four different step sizes, the deterministic artificial intelligence approach shows more tracking errors than the model-following approach when using larger sampling periods (faster computation rates). As shown in Table 3, this phenomenon becomes exacerbated with even a slight step size increase. It is obvious from observing Figure 4 below that there is a discrepancy between those two approaches at a large step size. The output signal from deterministic artificial intelligence shows a large oscillation at the beginning, while the model-following output signal tracks the command signal immediately.

**Table 3.** Error distribution of comparison between model-following control (MF) and deterministic artificial intelligence with different step sizes.

| Method | Step Size (s) | Error Mean | Error Standard Deviation |
|---|---|---|---|
| Deterministic artificial intelligence | 0.60 | 6.3918 | 4.8100 |
| Model-following | 0.60 | 0.0264 | 0.0973 |
| Deterministic artificial intelligence | 0.50 | 0.0956 | 0.1632 |
| Model-following | 0.50 | 0.0277 | 0.0917 |
| Deterministic artificial intelligence | 0.27 | 0.0175 | 0.0545 |
| Model-following | 0.27 | 0.0471 | 0.1745 |
| Deterministic artificial intelligence | 0.20 | 0.0114 | 0.0487 |
| Model-following | 0.20 | 0.0608 | 0.2446 |

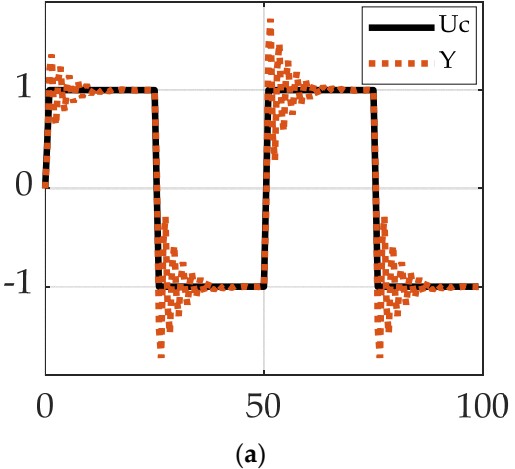
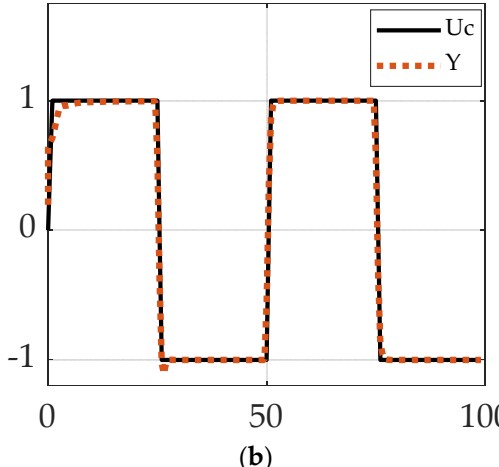

(**a**)  (**b**)

**Figure 4.** The output signal for the control approaches with a 0.5 s step size. The black line represents the command signal. (**a**) Output signal obtained from discrete deterministic artificial intelligence approach. (**b**) Output signal generated from continuous deterministic artificial intelligence approach.

According to Table 3, when the step size increases from 0.5 to 0.6 s, the mean tracking error rises sharply, becoming 242 times the mean error of the model-following approach. The standard deviation also increases dramatically, which is 49 times the standard deviation of the model-following method.

However, the performance becomes significantly more reliable when the step sizes become smaller. When the step size is reduced to 0.20 s, the mean error of the deterministic artificial intelligence approach decreases to 10% of it with a 0.5 s step size. The standard deviation also reduces to 1/3 of it with a 0.5 s step size. For the output of deterministic artificial intelligence, there is an overshoot at discontinuous, which follows the input signal with a small tracking error, which Koo [13] also mentions. In contrast, there is a degradation in the performance of the model-following approach when the step size is reduced from 0.6 to 0.2 s. An oscillation can be observed as the step size becomes smaller.

*3.2. Discrete Deterministic Artificial Intelligence vs. Continuous Deterministic Artificial Intelligence*

Table 4 shows the comparison of the error distributions between discrete deterministic artificial intelligence and continuous deterministic artificial intelligence. Overall, as already discussed in the Materials and Methods section, deterministic artificial intelligence has less efficacy with a large step size, which can be observed below. When the step size changes from 0.5 to 0.2 s, discrete and continuous deterministic artificial intelligence performance increases significantly. According to the error mean in Table 4, the mean error between

those two deterministic artificial intelligence is close to each other with a small step size. The difference is just about 7%.

**Table 4.** Error distribution of comparison between discrete deterministic artificial intelligence and continuous deterministic artificial intelligence with different step sizes [1].

| Deterministic Artificial Intelligence Type | Step Size (s) | Error Mean | Error Standard Deviation |
|---|---|---|---|
| Discrete | 0.50 | 0.0956 | 0.1632 |
| Continuous | 0.50 | 0.0223 | 0.1654 |
| Discrete | 0.20 | 0.0114 | 0.0487 |
| Continuous | 0.20 | 0.0122 | 0.1401 |

[1] Integration solver step size matched to discretization interval.

Although the error standard deviation of discrete deterministic artificial intelligence is half that of continuous deterministic artificial intelligence, a minor standard deviation does not represent better performance. The larger standard deviation of continuous deterministic artificial intelligence is caused by the more prominent initial oscillation in the initial transient, shown in Figure 5. After the oscillation section, the performance of continuous deterministic artificial intelligence is expected to exceed the discrete deterministic artificial intelligence since it has a marginal tracking error. MATLAB® obtains the results of Section 3. The code is included in Appendix A to help with replications or further constructions based on this article.

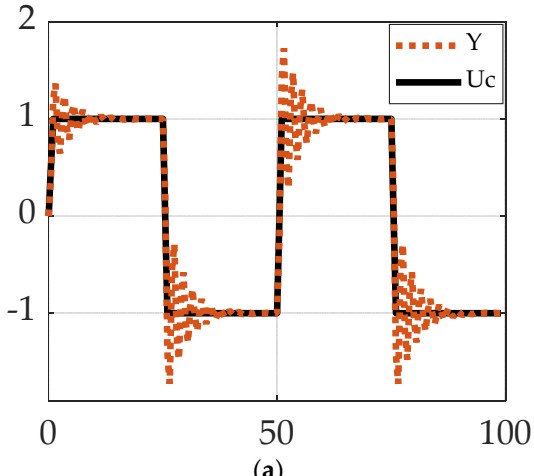

(a)

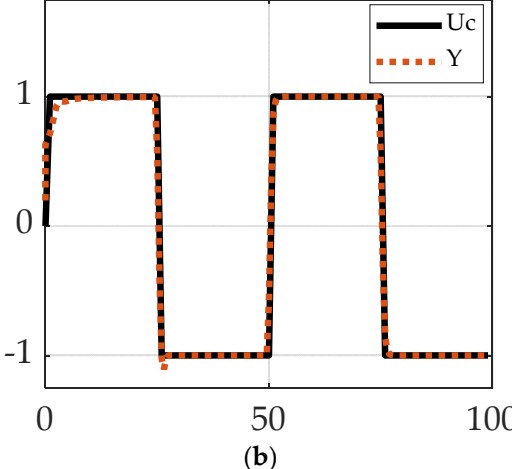

(b)

**Figure 5.** The output signal for the deterministic artificial intelligence approaches with a 0.5 s step size. The black line represents the command signal. (**a**) Output signal obtained from discrete deterministic artificial intelligence approach. (**b**) Output signal generated from continuous deterministic artificial intelligence approach.

## 4. Discussion

Tables 5 and 6 show the performance improvement in different algorithms. This manuscript validates the deterministic artificial intelligence approaches ability to track the discontinuous command square wave compared with alternative techniques. The exemplary performance of the continuous control is examined. Further discretization could be applied to obtain a better performance for the continuous control. As described in Table 3 of [13], different discretization methods, such as the zero-order hold (ZOH), bilinear approximation (Tustin), and linear interpolation (FOH), could significantly reduce the tracking error with large step sizes.

**Table 5.** Performance improvement with different step sizes [1].

| Method | Step Size (s) | Error Mean | Error Standard Deviation |
|---|---|---|---|
| DAI | 0.60 | 6586% | 2847% |
| MF | 0.60 | −72% | −40% |
| DAI | 0.50 | 0% | 0% |
| MF | 0.50 | −71% | −44% |
| DAI | 0.27 | −82% | −67% |
| MF | 0.27 | −51% | 7% |
| DAI | 0.20 | −88% | −70% |
| MF | 0.20 | −36% | 50% |

[1] Model-following (MF) control and deterministic artificial intelligence (DAI).

**Table 6.** Performance improvement for discrete and continuous deterministic artificial intelligence with different step sizes.

| Deterministic Artificial Intelligence Type | Step Size (s) | Error Mean | Error Standard Deviation |
|---|---|---|---|
| Discrete | 0.50 | 0% | 0% |
| Continuous | 0.50 | −77% | 1% |
| Discrete | 0.20 | −88% | −70% |
| Continuous | 0.20 | −87% | −14% |

As shown in Figures 4 and 5, this manuscript included the control effects for different control algorithms. Moreover, the performance of different algorithms at disparate step sizes is also compared in previous tables and figures. As further research of the article written by Koo [13], more step sizes of different control methods were applied in this manuscript to further validate the deterministic artificial intelligence performance over various situations. Overall, the discrepancy of the output signal from the different algorithms decreased when the step size reduced. Eventually, it became negligible. Discrete deterministic artificial intelligence seems to have the best performance under a slow computation rate (small step sizes), and continuous deterministic artificial intelligence is next. However, since continuous deterministic artificial intelligence is superior with various computation rate compared with other approaches overall, it might be the most feasible and reliable selection for future applications on unmanned underwater vehicles.

The future research recommendations are to replicate all the benchmarks for the sequel study and validate more parameters influenced by different control algorithms. Given these promising results, it is important to continue researching and refining deterministic artificial intelligence to improve its performance further. In particular, future research should focus on eliminating the sole transient that occurs on the initial startup, as this could lead to even more accurate and stable estimates. Many potential approaches could be used to address this issue, including the development of new computational techniques, the incorporation of additional data or model assumptions, and the use of different types of algorithms or models. By carefully exploring these options, it may be possible to develop deterministic artificial intelligence that is even more effective at estimating parameters in various situations.

### 4.1. Concluding Remarks

This manuscript demonstrates the performance of deterministic artificial intelligence and the model-following control algorithms on DC motor control over multiple computation rates, which is inversely proportional to the step size. The discrepancy of the output signal from the different algorithms decreased when the step size was reduced and eventually became negligible. Although discrete deterministic artificial intelligence behaves satisfactorily under a slow computation rate, continuous deterministic artificial intelligence is superior with various computation rates. Overall, continuous deterministic artificial

intelligence is the most feasible and reliable selection for future applications on unmanned underwater vehicles.

*4.2. Recommended Future Research*

Continuing the lineage of this research's efforts, the next logical steps to investigate include replicating all the new benchmarks in a sequel study and validating more parameters that are influenced by different control algorithms. Furthermore, an effective technique that can eliminate the initial interruption caused by the initial transient will improve the accuracy of the control algorithm, which should seemingly significantly increase the performance.

**Author Contributions:** Conceptualization, Z.W. and T.S.; methodology, Z.W. and T.S.; software, Z.W. and T.S.; validation, T.S.; formal analysis, Z.W. and T.S.; investigation, Z.W. and T.S.; resources, T.S; writing—original draft preparation, Z.W.; writing—review and editing, Z.W. and T.S.; supervision, T.S.; funding acquisition, T.S. All authors have read and agreed to the published version of the manuscript.

**Funding:** This research received no external funding.

**Data Availability Statement:** Data supporting reported results can be obtained by contacting the corresponding author.

**Acknowledgments:** The code used to generate data and figures in this manuscript is adapted from the Appendix A of Koo Mo Koo and Henry Travis's article [13].

**Conflicts of Interest:** The authors declare no conflict of interest.

## Appendix A

The appendix contains topologies that are crucial to understanding and reproducing the research published in this manuscript. The code should be run by MATLAB®.

*Appendix A.1. Discrete Deterministic Artificial Intelligence*

```
clear all; clc; close all;
%% DISCRETIZATION
% B = [0 0.1065 0.0902]; A = poly([1.1 0.8]);
% Gs = tf(B,A);
% a1 = 0; a2 = 0; b0= 0.1; b1 = 0.2; %Shah's
Bp = [0 0 1];
Ap = [1 1 0];
Gs = tf(Bp,Ap); %Create continuous time transfer function
Ts = 0.5;
Hd = c2d(Gs,Ts,'matched'); % Transform continuous system to discrete system
B = Hd.Numerator{1};
A = Hd.Denominator{1};
b0 = 0.1; b1 = 0.1; a0 = 0.1; a1 = 0.01; a2 = 0.01;
%% RLS
Am = poly([0.2+0.2j 0.2-0.2j]);
Bm = [0 0.1065 0.0902];
am0 = Am(1); am1 = Am(2); am2 = Am(3); a0 = 0;
Rmat = [];
factor = 25;
% Reference
T_ref = 25;
t_max = 100;
time = 0:0.5:t_max;
nt = length(time);
% slew stuff
```

```
Tslew = 1;
Uc = zeros(length(nt));
for j = 1:nt
% pos or neg
if mod(time(j),2*T_ref) < T_ref
pn = 1;
else
pn =−1;
end
% slew
if mod(time(j),T_ref) < Tslew
Uc(j) = pn*-1*sin(pi/2+pi/Tslew*mod(time(j),T_ref));
else
Uc(j) = pn;
end
% initial slew special case
if time(j) < Tslew
Uc(j) = 1/2*-1*sin(pi/2+pi/Tslew*mod(time(j),T_ref)) + 1/2;
end
end
n = 4;lambda = 1.0;
nzeros = 2;
time = zeros(1,nzeros);
Y = zeros(1,nzeros);
Ym = zeros(1,nzeros);
U=ones(1,nzeros);
Uc=[zeros(1,nzeros),Uc];
Noise = 0;
P = [100 0 0 0;0 100 0 0;0 0 1 0;0 0 0 1];
THETA_hat(:,1) = [-a1 -a2 b0 b1]';
beta = [];
alpha = 0.5;
gamma = 1.2;
for i = 1:201
phi = [];
t = i+nzeros;
time(t ) = i;
Y(t) = [-A(2) -A(3) B(2) B(3)]*[Y(t-1) Y(t-2) U(t-1) U(t-2)]';
Ym(t) = [-Am(2) -Am(3) Bm(2) Bm(3)]*[Ym(t-1) Ym(t-2) Uc(t-1) Uc(t-2)]';
BETA = (Am(1)+Am(2)+Am(3))/(b0+b1);
beta = [beta BETA];
%RLS implementation
phi = [Y(t-1) Y(t-2) U(t-1) U(t-2)]';
K = P*phi*1/(lambda+phi'*P*phi);
P=P-P*phi*inv(1+phi'*P*phi)*phi'*P/lambda; %RLS-EF
error(i) = Y(t)-phi'*THETA_hat(:,i);
THETA_hat(:,i+1) = THETA_hat(:,i)+K*error(i);
a1 = -THETA_hat(1,i+1);
a2 = -THETA_hat(2,i+1);
b0 = THETA_hat(3,i+1);
b1 = THETA_hat(4,i+1);
Af(:,i) = [1 a1 a2]';
Bf(:,i) = [b0 b1]';
% Determine R,S, & T for CONTROLLER
```

```
r1 = (b1/b0)+(b1^2-am1*b0*b1+am2*b0^2)*(-b1+a0*b0)/(b0*(b1^2-a1*b0*b1+a2*b0^2));
s0 = b1*(a0*am1-a2-am1*a1+a1^2+am2-a1*a0)/(b1^2-a1*b0*b1+a2*b0^2)+b0*(am1*
a2-a1*a2-a0*am2+a0*a2)/(b1^2-a1*b0*b1+a2*b0^2);
s1 = b1*(a1*a2-am1*a2+a0*am2-a0*a2)/(b1^2-a1*b0*b1+a2*b0^2)+b0*(a2*am2-a2^2-a0*
am2*a1+a0*a2*am1)/(b1^2-a1*b0*b1+a2*b0^2);
R = [1 r1];
S = [s0 s1];
T = BETA*[1 a0];
Rmat = [Rmat r1];
%calculate control signal
U(t) = [T(1) T(2) -R(2) -S(1) -S(2)]*[Uc(t) Uc(t-1) U(t-1) Y(t) Y(t-1)]';
U(t) = 1.3*[T(1) T(2) -R(2) -S(1) -S(2)]*[Uc(t) Uc(t-1) U(t-1) Y(t) Y(t-1)]';% Arbitrarily
increased to duplicate text
end
%% deterministic artificial intelligence
%Create command signal, Uc based on Example 3.5 plots . . . square wave with 50 sec
period
t_max = 200;
THETA_hat(:,1) = [-a1 -a2 b0 b1]';
n = length(THETA_hat);
% Sigma = 1/25; Noise=Sigma*randn(nt,1);
% Noise = 0;
nzeros = 2;
Y_true = zeros(1,nzeros);
Ym = zeros(1,nzeros);
U = zeros(1,nzeros);
P = [100 0 0 0;0 100 0 0;0 0 1 0;0 0 0 1];
lambda = 1;
eb = Y_true(1) - Uc(1);
err = 0;
kp = 2.0;
kd = 6.0;
hatvec = zeros(4,1);
for i = 1:t_max+1 %Loop through the output data Y(t)
t = i+nzeros;
de = err-eb;
u = kp*err + kd*de;
U(t-1) = u;
Y_true(t) = [Y_true(t-1) Y_true(t-2) U(t-1) U(t-2)]*[-A(2) -A(3) B(2) B(3)]';
phid = [Y_true(t) -Y_true(t-1) Y_true(t-2) -U(t-2)];
newest = phid\u;
hatvec(:,i) = newest;
eb = err;
%disp(t);
err = Uc(t)-Y_true(t);
end
%% PLOT
tspan = linspace(0,100,201);
tspan = [zeros(1,2) tspan];
figure(1); %deterministic artificial intelligence
plot(tspan(1:201),Uc(1:201));
hold on;
plot(tspan(1:201),Y_true(2:202));
hold off
```

```
xlabel('Time(sec)');
legend('Uc','Y','fontsize',11);
set(gca,'fontsize',16);
set(gca,'fontname','Palatino Linotype');
xlim([0 max(time)]); grid;
% p = plot(tspan,Uc(1:203),'-',tspan,Y,'-');
% p(2).LineWidth = 2;
% legend('Uc','Y','fontsize',11);
%deterministic artificial intelligence
axis([0 100,-1.5 1.5]);
figure(2); %RLS estimation
plot(tspan(1:201),Uc(1:201),'k-','LineWidth',1);
hold on;
plot(tspan(1:201),Y(3:203));
hold off
xlabel('Time(sec)');
legend('Uc','Y','fontsize',11);
set(gca,'fontsize',16);
set(gca,'fontname','Palatino Linotype');
xlim([0 max(time)]);
grid;
axis([0 100,-1.5 1.5]);
deterministic artificial intelligence_err_mean = mean(abs(Uc(1:201)-Y_true(2:202)))
deterministic artificial intelligence_err_std = std(abs(Uc(1:201)-Y_true(2:202)))
RLS_err_mean = mean(abs(Uc(1:201)-Y(3:203)))
RLS_err_std = std(abs(Uc(1:201)-Y(3:203)))
```

*Appendix A.2. Continuous Deterministic Artificial Intelligence*

```
clear all;clc;close all;
% Enter Given Plant parameters
for k = 1:2
Bp = [0 0 1];
Ap = [1 1 0];
Gs = tf(Bp,Ap); %Create continuous time transfer function
Ts = [0.5 0.2];
Hz = c2d(Gs,Ts(k),'matched'); % Transform continuous system to discrete system
B = Hz.Numerator{1};
A = Hz.Denominator{1};
% Initial estimates of plant parameters for undetermined system from example 3.5
b0 = 0.1; b1 = 0.1; a0 = 0.1; a1 = 0.01; a2 = 0.01;
% Reference
T_ref = 25;
t_max = 100;
time = 0:Ts:t_max;
nt = length(time);
% slew stuff
Tslew = 1;
Yd = zeros(length(nt));
for i=1:nt
% pos or neg
if mod(time(i),2*T_ref) < T_ref
pn = 1;
else
pn = -1;
```

```
end
% slew
if mod(time(i),T_ref)<Tslew
Yd(i) = pn*-1*sin(pi/2+pi/Tslew*mod(time(i),T_ref));
else
Yd(i) = pn;
end
% initial slew special case
if time(i) < Tslew
Yd(i) = 1/2*-1*sin(pi/2+pi/Tslew*mod(time(i),T_ref))+1/2;
end
end
THETA_hat(:,1) = [-a1 -a2 b0 b1]';
n = length(THETA_hat);
Sigma = 1/12*0;
Noise = Sigma*randn(nt,1);
nzeros = 2;
Y = zeros(1,nzeros);
Y_true = zeros(1,nzeros);
Ym = zeros(1,nzeros);
U = zeros(1,nzeros);
Yd = [zeros(1,nzeros),Yd];
P = [100 0 0 0;0 100 0 0;0 0 1 0;0 0 0 1];
lambda = 1;
for i = 1:nt-1
t = i+nzeros;
% Update Dynamics
Y_true(t) = [Y(t-1) Y(t-2) U(t-1) U(t-2)]*[-A(2) -A(3) B(2) B(3)]';
Y(t) = Y_true(t)+Noise(i);
phi = [Y(t-1) Y(t-2) U(t-1) U(t-2)]';
K = P*phi*1/(lambda+phi'*P*phi);
P = P-P*phi/(1+phi'*P*phi)*phi'*P/lambda;
innov_err(i) = Y(t)-phi'*THETA_hat(:,i);
THETA_hat(:,i+1) = THETA_hat(:,i)+K*innov_err(i);
a1 = -THETA_hat(1,i+1);a2 = -THETA_hat(2,i+1);
b0 = THETA_hat(3,i+1);b1 = THETA_hat(4,i+1);% THETA = [-a1 -a2 b0 b1];
% Calculate Model control, U(t) optimally
U(t) = [Yd(t+1) Y(t) Y(t-1) U(t-1)]*[1 a1 a2 -b0]'/b1;
end
Y_true(end+1) = Y_true(end);
FS = 2;
time = [-(nzeros-1)*Ts:Ts:0 time];
figure (k)
plot(time,Yd,'k-','LineWidth',2);
hold on;
h1 = plot(time,Y_true,'LineWidth',1);
axis([0 100,-1.5 1.5]);
hold off;
grid;
if k==1
legend(h1,'T_s = 0.50s','fontsize',11);
xlabel('Time(sec)');
set(gca,'fontsize',16);
set(gca,'fontname','Palatino Linotype');
```

```
else
legend(h1,'T_s = 0.20s','fontsize',11);
xlabel('Time(sec)');
set(gca,'fontsize',16);
set(gca,'fontname','Palatino Linotype');
end
end
```

*Appendix A.3. All Deterministic Artificial Intelligence*

```
clear all;clc;close all;
%% DISCRETIZATION
% B = [0 0.1065 0.0902]; A = poly([1.1 0.8]);
% Gs = tf(B,A);
% a1 = 0;a2 = 0;b0 = 0.1;b1 = 0.2; %Shah's
Ap = [1 1 0];
Bp = [0 0 1];
Gs = tf(Bp,Ap); %Create continuous time transfer function
Ts = 0.5;
Hd = c2d(Gs,Ts,'matched'); % Transform continuous system to discrete system
B = Hd.Numerator{1};
A = Hd.Denominator{1};
b0 = 0.1; b1 = 0.1;
a1 = 0.01; a2 = 0.01;
%% RLS
Am = poly([0.2+0.2j 0.2-0.2j]);
Bm = [0 0.1065 0.0902];
am0 = Am(1);am1 = Am(2);am2 = Am(3);a0 = 0;
Rmat = [];
factor = 25;
% Reference
T_ref = 25;
t_max = 100;
time = 0:0.5:t_max;
nt = length(time);
% slew stuff
Tslew = 1;
Uc = zeros(length(nt));
for j = 1:nt
% pos or neg
if mod(time(j),2*T_ref)<T_ref
pn = 1;
else
pn =-1;
end
% slew
if mod(time(j),T_ref)<Tslew
Uc(j) = pn*-1*sin(pi/2+pi/Tslew*mod(time(j),T_ref));
else
Uc(j) = pn;
end
% initial slew special case
if time(j)<Tslew
Uc(j) = 1/2*-1*sin(pi/2+pi/Tslew*mod(time(j),T_ref))+1/2;
end
```

```
end
n = 4;
lambda = 1.0;
nzeros = 2;
time = zeros(1,nzeros);
Y = zeros(1,nzeros);
Ym = zeros(1,nzeros);
U = ones(1,nzeros);
Uc = [zeros(1,nzeros),Uc];
Noise = 0;
P = [100 0 0 0;0 100 0 0;0 0 1 0;0 0 0 1];
THETA_hat(:,1) = [-a1 -a2 b0 b1]';
beta = [];
alpha = 0.5;
gamma = 1.2;
for i = 1:201
phi = [];
t = i+nzeros;
time(t) = i;
Y(t) = [-A(2) -A(3) B(2) B(3)]*[Y(t-1) Y(t-2) U(t-1) U(t-2)]';
Ym(t) = [-Am(2) -Am(3) Bm(2) Bm(3)]*[Ym(t-1) Ym(t-2) Uc(t-1) Uc(t-2)]';
BETA = (Am(1)+Am(2)+Am(3))/(b0+b1);
beta = [beta BETA];
%RLS implementation
phi = [Y(t-1) Y(t-2) U(t-1) U(t-2)]';
K = P*phi*1/(lambda+phi'*P*phi);
P = P-P*phi*inv(1+phi'*P*phi)*phi'*P/lambda; %RLS-EF
error(i) = Y(t)-phi'*THETA_hat(:,i);
THETA_hat(:,i+1) = THETA_hat(:,i)+K*error(i);
a1 = -THETA_hat(1,i+1);a2 = -THETA_hat(2,i+1);
b0 = THETA_hat(3,i+1);b1 = THETA_hat(4,i+1);
Af(:,i) = [1 a1 a2]';
Bf(:,i) = [b0 b1]';
% Determine R,S, & T for CONTROLLER
r1 = (b1/b0)+(b1^2-am1*b0*b1+am2*b0^2)*(-b1+a0*b0)/(b0*(b1^2-a1*b0*b1+a2*b0^2));
s0 = b1*(a0*am1-a2-am1*a1+a1^2+am2-a1*a0)/(b1^2-a1*b0*b1+a2*b0^2)+b0*(am1*
a2-a1*a2-a0*am2+a0*a2)/(b1^2-a1*b0*b1+a2*b0^2);
s1 = b1*(a1*a2-am1*a2+a0*am2-a0*a2)/(b1^2-a1*b0*b1+a2*b0^2)+b0*(a2*am2-a2^2-a0*
am2*a1+a0*a2*am1)/(b1^2-a1*b0*b1+a2*b0^2);
R = [1 r1];
S = [s0 s1];
T = BETA*[1 a0];
Rmat = [Rmat r1];
%calculate control signal
U(t) = [T(1) T(2) -R(2) -S(1) -S(2)]*[Uc(t) Uc(t-1) U(t-1) Y(t) Y(t-1)]';
U(t) = 1.3*[T(1) T(2) -R(2) -S(1) -S(2)]*[Uc(t) Uc(t-1) U(t-1) Y(t) Y(t-1)]';% Arbitrarily
increased to duplicate text
end
%% deterministic artificial intelligence
%Create command signal, Uc based on Example 3.5 plots . . . square wave with 50 sec
period
t_max = 200;
THETA_hat(:,1) = [-a1 -a2 b0 b1]';
n = length(THETA_hat);
```

```
% Sigma = 1/25; Noise = Sigma*randn(nt,1);
% Noise = 0;
nzeros = 2;
Y_true = zeros(1,nzeros);
Ym = zeros(1,nzeros);
U = zeros(1,nzeros);
P = [100 0 0 0;0 100 0 0;0 0 1 0;0 0 0 1];
lambda = 1;
eb = Y_true(1) - Uc(1);
err = 0;
kp = 2.0;
kd = 6.0;
hatvec = zeros(4,1);
for i = 1:t_max+1 %Loop through the output data Y(t)
t = i+nzeros;
de = err-eb;
u = kp*err + kd*de;
U(t-1) = u;
Y_true(t) = [Y_true(t-1) Y_true(t-2) U(t-1) U(t-2)]*[-A(2) -A(3) B(2) B(3)]';
phid = [Y_true(t) -Y_true(t-1) Y_true(t-2) -U(t-2)];
newest = phid\u
hatvec(:,i) = newest;
eb = err;
%disp(t);
err = Uc(t)-Y_true(t);
end
%%% PLOT
tspan = linspace(0,100,201);
tspan = [zeros(1,2) tspan];
figure(1); %deterministic artificial intelligence
plot(tspan(1:201),Uc(1:201),'k-','LineWidth',1);
hold on;
plot(tspan(1:201),Y_true(2:202),'LineWidth',3);
hold off
xlabel('Time(sec)');
legend('Uc','Y','fontsize',11);
set(gca,'fontsize',16);
set(gca,'fontname','Palatino Linotype');
xlim([0 max(time)]);
grid;
% p = plot(tspan,Uc(1:203),'-',tspan,Y,'-'); p(2).LineWidth = 2; legend('Uc', 'Y', 'font-
size',11);
%deterministic artificial intelligence
axis([0 100,-1.5 1.5]);
figure(2); %RLS estimation
plot(tspan(1:201),Uc(1:201),'k-','LineWidth',1);
hold on;
plot(tspan(1:201),Y(3:203),'LineWidth',3);
hold off
xlabel('Time(sec)');
legend('Uc','Y','fontsize',11);
set(gca,'fontsize',16);
set(gca,'fontname','Palatino Linotype');
xlim([0 max(time)]); grid;
```

axis([0 100,-1.5 1.5]);
deterministic artificial intelligence_err_mean = mean(abs(Uc(1:201)-Y_true(2:202)))
deterministic artificial intelligence_err_std = std(abs(Uc(1:201)-Y_true(2:202)))
RLS_err_mean = mean(abs(Uc(1:201)-Y(3:203)))
RLS_err_std = std(abs(Uc(1:201)-Y(3:203)))

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
