# Peer review of "Artificial Intelligence-Enhanced UUV Actuator Control"

_ai, doi:10.3390/ai4010012_

Round 1

Reviewer 1 Report

The idea is interesting but the following problems should be considered during the preparation of the revised version.

1) The abstract should be clear with some information about the type of AI used in the paper.

2) The literature review considering the problem and also AI part should be improved.

3) Using some pseudocode may improve the readability of the paper.

4) The appendix should be placed after references.

5) The paper needs proofreading.

Author Response

Reviewer 1, Thanks for all the great recommendations. All revision requests were ceded without argument, and the ameliorating revisions embody an increase in manuscript length of 8.5%.

The idea is interesting but the following problems should be considered during the preparation of the revised version.

1) The abstract should be clear with some information about the type of AI used in the paper.

Thank you. The abstract now articulates the type of AI five times in 14 lines.

2) The literature review considering the problem and also AI part should be improved.

Thank you. The literature review has been augmented with the key citation of the development of DAI and also requisite references about the system parameterization.

3) Using some pseudocode may improve the readability of the paper.

Thank you. The code is now included in the Appendix A to aid the repeatability of the research by the readership.

4) The appendix should be placed after references.

Thank you. The references section is moved before the appendix.

5) The paper needs proofreading.

Thank you. Grammar mistakes are eliminated, and the sentences are updated.

Reviewer 2 Report

The current manuscript compares deterministic artificial intelligence to model following control applied to DC motor control, including evaluation of the threshold computation rate to let un-manned underwater vehicles correctly follow the challenging discontinuous square wave command signal. This reviewer thinks the current work needs more attention with the following issues: 

1. Keywords: use the most efficient 6 keywords.

2. Introduction section: Add the research gap and authors contribution.  Using figures is not perfereable in this section. Extend the survey to more updated references. Don't use many references without critcal reviews for each as in [31]-[37]. Reaarange of this section is needed to show the motivation, literature review, research gap, contribution and orginization.

3. Description of the deterministic artificial intelligence and continous deterministic AI needs more work with aid of flow chart and numerical assessment.

4. Discussion section is limited. 

5. The work has not any conclusion remarks.

This study needs more work and reorginization.

3. 

3.

Author Response

Reviewer 2, Thanks for all the great recommendations. All revision requests were ceded without argument, and the ameliorating revisions embody an increase in manuscript length of 8.5%.

The current manuscript compares deterministic artificial intelligence to model following control applied to DC motor control, including evaluation of the threshold computation rate to let un-manned underwater vehicles correctly follow the challenging discontinuous square wave command signal. This reviewer thinks the current work needs more attention with the following issues:

  1. Keywords: use the most efficient 6 keywords.

- Thank you.  The keywords have been reduced to six.

  1. Introduction section:
  2. Add the research gap and authors contribution.
  • Thank you. Reviewer’s request is ceded and ameliorated in the revision.
  1. Using figures is not perfereable in this section.
  • Thank you. We identify inclusion on the first page as a “best practice” in professional technical writing, especially when the image well pairs with the large bolded article title. A decent discussion is available here: https://courses.lumenlearning.com/suny-jeffersoncc-technicalwriting/chapter/photos-and-illustrations/.  Best practices are included here: https://www.gcu.edu/blog/criminal-justice-government-and-public-administration/best-practices-technical-writing  
  1. Extend the survey to more updated references.
  • The updated survey has additional recent references raising the ratio of recent references to the historical lineage to over 55%.
  1. Don't use many references without critcal reviews for each as in [31]-[37].
  • In the revision, only [31,32] have grouped descriptions as mere evaluation of feedforward elements, while [33-37] are described as evaluations of separate individual efficacies.
  1. Reaarange of this section is needed to show the motivation, literature review, research gap, contribution and orginization.
  • The recommendation is ceded and accommodated in the revision.
  1. Description of the deterministic artificial intelligence and continous deterministic AI needs more work with aid of flow chart and numerical assessment.
  • The recommendation is ceded and accommodated in the revision.
  1. Discussion section is limited.
  • The recommendation is ceded and accommodated in the revision. Since the discussion is not particularly long nor complicated, concluding remarks are appended to the discussion section in accordance with MDPI recommended practices.
  1. The work has not any conclusion remarks.
  • The recommendation is ceded and accommodated in the revision’s newly structured discussion.
  1. This study needs more work and reorginization.
  • The recommendation is ceded and accommodated in the revision with several structural changes and improved organization.

Grammar mistakes are eliminated, and the sentences are updated. Sections are reorganized.

Reviewer 3 Report

The topic of the paper is very interesting and of high interest to potential readers. However, I find that in its present form, the paper does not do justice to the authors. 

The introduction needs, in my opinion, re-writing, in order to highlight in a better way the limitations of existing related work that this paper aims to address. Also a brief explanation of the problem and its aspects would increase the paper's accessibility to readers that are not already too familiar with the topic. (what is discrete AI, why is it important to reduce error etc). Most equations do not have sufficient explanation of what the parameters that they include refer to, and this is an area that needs improvement.

An overall check of the language is also necessary because some sentences seem to be missing verbs or have very strange syntax. 

Author Response

Reviewer 3Thanks for all the great recommendations. All revision requests were ceded without argument, and the ameliorating revisions embody an increase in manuscript length of 8.5%.

The topic of the paper is very interesting and of high interest to potential readers. However, I find that in its present form, the paper does not do justice to the authors.

  1. The introduction needs, in my opinion, re-writing, in order to highlight in a better way the limitations of existing related work that this paper aims to address.
  • The recommendation is ceded and accommodated in the revision’s new figure 2 placed amidst the respective cited literature establishing the research lineage.
  1. Also a brief explanation of the problem and its aspects would increase the paper's accessibility to readers that are not already too familiar with the topic. (what is discrete AI, why is it important to reduce error etc).
  • The recommendation is ceded and accommodated with the new flow chart.
  1. Most equations do not have sufficient explanation of what the parameters that they include refer to, and this is an area that needs improvement.
  • Periodic tables of proximal variables and nomenclature are added throughout the manuscript to aid readability.
  1. An overall check of the language is also necessary because some sentences seem to be missing verbs or have very strange syntax.

Grammar mistakes are eliminated, and the sentences are updated. Sections are reorganized.

Round 2

Reviewer 1 Report

All of my concerns have been addressed.

Reviewer 2 Report

No other comments